# LRA-QViT: Integrating Low-Rank Approximation and Quantization for Robust and Efficient Vision Transformers

Beom Jin Kang [1]  Nam Joon Kim [1]  Hyun Kim [1]

## Abstract

Recently, transformer-based models have demonstrated state-of-the-art performance across various computer vision tasks, including image classification, detection, and segmentation. However, their substantial parameter count poses significant challenges for deployment in resource-constrained environments such as edge or mobile devices. Low-rank approximation (LRA) has emerged as a promising model compression technique, effectively reducing the number of parameters in transformer models by decomposing high-dimensional weight matrices into low-rank representations. Nevertheless, matrix decomposition inherently introduces information loss, often leading to a decline in model accuracy. Furthermore, existing studies on LRA largely overlook the quantization process, which is a critical step in deploying practical vision transformer (ViT) models. To address these challenges, we propose a robust LRA framework that preserves weight information after matrix decomposition and incorporates quantization tailored to LRA characteristics. First, we introduce a reparameterizable branch-based low-rank approximation (RB-LRA) method coupled with weight reconstruction to minimize information loss during matrix decomposition. Subsequently, we enhance model accuracy by integrating RB-LRA with knowledge distillation techniques. Lastly, we present an LRA-aware quantization method designed to mitigate the large outliers generated by LRA, thereby improving the robustness of the quantized model. To validate the effectiveness of our approach, we conducted extensive experiments on the ImageNet dataset

using various ViT-based models. Notably, the Swin-B model with RB-LRA achieved a 31.8% reduction in parameters and a 30.4% reduction in GFLOPs, with only a 0.03% drop in accuracy. Furthermore, incorporating the proposed LRA-aware quantization method reduced accuracy loss by an additional 0.83% compared to naive quantization.

## 1. Introduction

Transformer-based models have demonstrated powerful performance in various fields, including computer vision, language processing, and speech processing (Vaswani et al., 2017; Dosovitskiy et al., 2020; Gulati et al., 2020). Specifically, vision transformer (ViT) models have achieved higher accuracy than convolutional neural networks (CNNs) (He et al., 2016; Choi et al., 2019; He et al., 2017) in vision tasks such as image classification (Touvron et al., 2021; Liu et al., 2021), detection (Carion et al., 2020), and segmentation (Strudel et al., 2021). However, ViTs typically involve more parameters and computational complexity than CNNs, making them challenging to deploy on mobile and edge devices with limited memory and computing resources, unlike high-performance GPU servers (Yin et al., 2022; Lee et al., 2024b). To address these challenges, various compression methods have been proposed, including quantization (Yuan et al., 2022), pruning (Jongho & Kim, 2024) and low-rank approximation (LRA) (Lee et al., 2024a; Kumar, 2022).

LRA is an effective method for reducing model size by decomposing weights into low-rank matrices (Hajimolahoseini et al., 2022; Zhang et al., 2023). However, applying LRA often results in a significant drop in accuracy due to the loss of weight information. Additionally, as the weights become shallower than those of the original large-scale model, it becomes challenging to recover the original accuracy using standard fine-tuning methods. To address these issues, previous studies have introduced fine-tuning methods based on knowledge distillation (KD) (Hinton et al., 2015). For example, (Guo et al., 2024) proposed a method that regularizes the perturbations caused by LRA while simultaneously applying KD to improve accuracy. (Yu & Wu, 2023) proposed a method that decomposes activations and determines the appropriate rank for each layer based on sensitivity to

[1]Department of Electrical and Information Engineering and Research Center for Electrical and Information Technology, Seoul National University of Science and Technology, Seoul, Republic of Korea. Correspondence to: Hyun Kim <hyunkim@seoultech.ac.kr>.

*Proceedings of the $42^{nd}$ International Conference on Machine Learning*, Vancouver, Canada. PMLR 267, 2025. Copyright 2025 by the author(s).

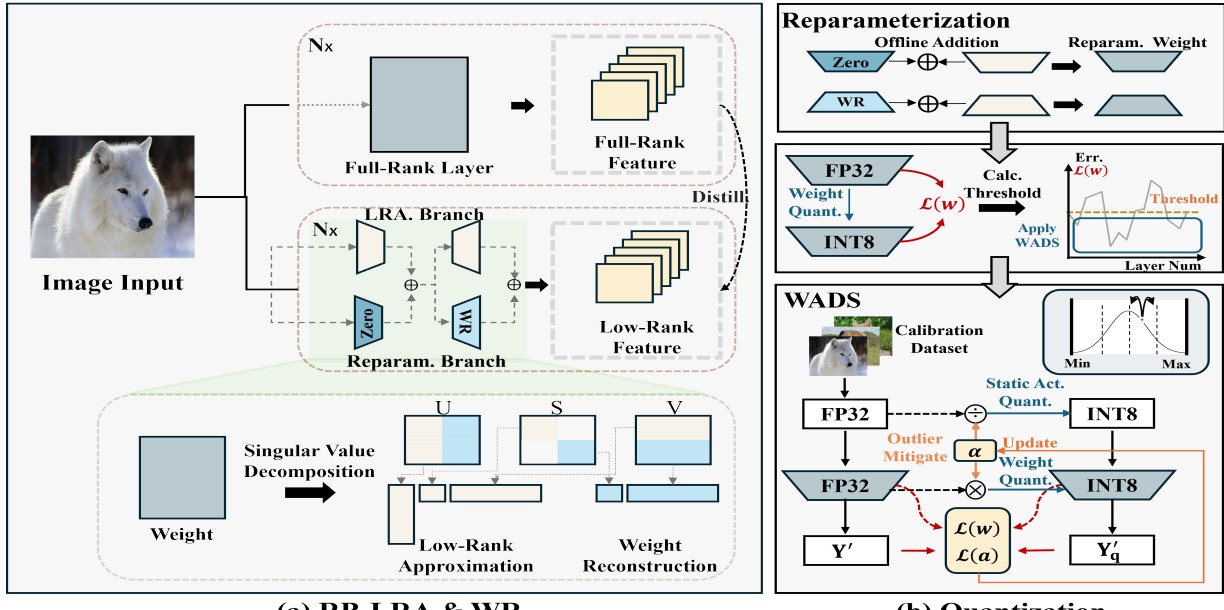

(a) RB-LRA & WR

(b) Quantization

*Figure 1.* The overall framework integrating the proposed RB-LRA method with quantization. The framework involves two steps: (a) Fine-tuning: Construct the RB-LRA layer by initializing the branch weights with discarded matrices from LRA, and apply feature-based knowledge distillation (KD) to improve performance. (b) Quantization: Create compressed weights by summing the weights of both the LRA and reparameterizable branches, then apply weight quantization and calculate the quantization error. Layers with errors below the threshold are processed with WADS to scale activations and apply quantization.

layer-wise perturbations. However, these approaches are suboptimal in minimizing information loss during weight decomposition, making it difficult to achieve high accuracy only using LRA. Therefore, to further improve the performance of LRA-based compression methods, a more robust LRA-oriented approach is needed.

Meanwhile, quantization reduces memory requirements and computational overhead by representing 32-bit floating point data as low-bit integers (*e.g.*, INT4, INT8). In resource-constrained environments, such as mobile and edge devices, quantization is essential for memory and power-efficient utilization of ViTs (Choi & Kim, 2022; Kang et al., 2024).

Integrating LRA and quantization methods offers an effective solution for the practical deployment of ViTs on mobile and edge devices, which are often constrained by limited computing and memory resources. Specifically, LRA can reduce the number of parameters and computational costs, while quantization enables integer-based operations that enhance inference speed and minimize memory usage. However, the simultaneous use of these two compression techniques can lead to severe accuracy degradation. In fact, we observed substantial outliers in specific channels and tokens after applying LRA, yet existing LRA studies fail to take quantization into consideration. Consequently, this observation highlights the need for a quantization solution that specifically addresses the effects of LRA.

Based on these motivations, we propose a framework that integrates robust LRA with quantization to efficiently utilize large ViT models on mobile and edge devices. Figure 1 illustrates the proposed integrated LRA and quantization framework. Our contributions are summarized as follows:

- We introduce a robust reparameterizable branch-based LRA (RB-LRA) and weight reconstruction (WR) methods that complement conventional LRA. The RB-LRA method employs a reparameterizable residual branch to compensate for LRA-induced errors. The WR method initializes the RB-LRA branch using the weights removed by LRA, thereby reducing information loss. Consequently, RB-LRA significantly mitigates accuracy degradation compared to existing LRA approaches.

- We observed large activation outliers when applying RB-LRA. To mitigate these outliers, we propose a weight-aware distribution scaling (WADS) method and suggest that per-token quantization is an effective approach to minimize accuracy drops caused by these outliers.

- We applied the proposed RB-LRA and WADS to representative ViT-based models (DeiT (Touvron et al., 2021), and Swin transformer (Liu et al., 2021)). Consequently, the proposed methods not only effectively

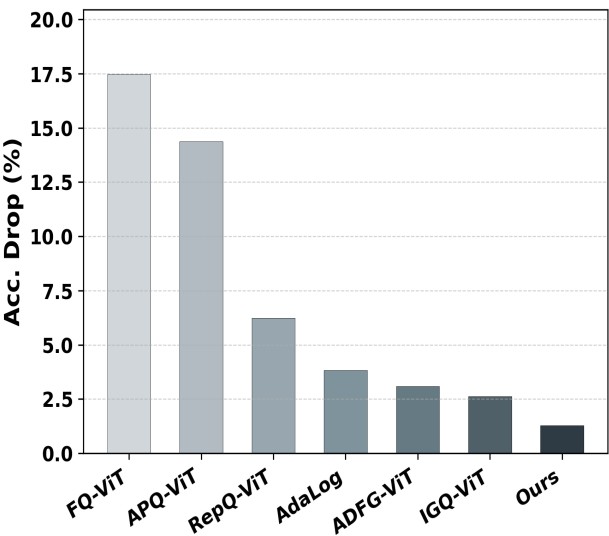

*Figure 2.* Comparison of Top-1 accuracy drop with various quantization methods and the our proposed compression framework combining LRA and quantization on the ImageNet dataset for the DeiT-B model. Detailed numerical results are provided in Appendix A.5.

enhance the trade-off between memory and accuracy but also achieve inference acceleration, making them well-suited for practical deployment on mobile and edge devices. Additionally, the proposed methods outperform existing quantization methods, as shown in Figure 2, when the DeiT-B model is compressed at a similar ratio (*i.e.,* INT4 quantization).

## 2. Related Works

### 2.1. Low-Rank Approximation

LRA is a method that decomposes a high-dimensional weight matrix into two low-rank matrices. It is primarily used to compress the fully connected (FC) layers, which are key components of transformer models. Among various methods, singular value decomposition (SVD)-based LRA (Klema & Laub, 1980) is one of the most frequently adopted approaches. SVD decomposes a matrix into a left singular vector matrix, a diagonal singular value matrix, and a right singular vector matrix. In this process, the matrix is approximated as a low-rank matrix by removing the smaller singular values while preserving the larger ones. However, removing these singular values leads to information loss, which causes a drop in accuracy. Accordingly, various methods have been proposed to address accuracy degradation. (Guo et al., 2024) proposed a method that integrates $l_\infty$-norm-based weight perturbation regularization with feature-based KD. (Yu & Wu, 2023) used an eigenvalue decomposition for decom-

posing the output feature map. However, (Guo et al., 2024) could not directly address the fundamental information loss that occurs during LRA. Instead, it focused on regularizing weight reconstruction perturbations and achieved improved performance by applying KD during fine-tuning. Moreover, (Yu & Wu, 2023) decomposed the output feature map rather than the weight matrix, also relying on KD. In contrast to these approaches, we propose a novel method that focuses on minimizing information loss through the reconstruction of weights removed during the LRA process.

### 2.2. Knowledge Distillation

Knowledge distillation (KD) is a method for distilling knowledge from a large model (*i.e.*, teacher model) to a smaller model (*i.e.*, student model). KD is typically categorized into two categories: response-based KD and feature-based KD. Response-based KD involves training the student model to replicate the probability distribution of the teacher model's output. For example, (Hinton et al., 2015) uses soft labels generated through label smoothing (Szegedy et al., 2016). This approach enhances generalization performance by training the student model on a probability distribution that spans all classes derived from the teacher's soft labels. In contrast, feature-based KD trains the student model to mimic the feature maps from specific layers or blocks of the teacher model. For instance, (Zagoruyko & Komodakis, 2016) distills attention map features from the teacher model into the student model. This approach aims to enhance performance by enabling the student model to replicate the feature distributions of the teacher model. Therefore, since LRA can significantly change the distribution of the original model, feature-based KD is an effective method to improve performance.

### 2.3. Quantization

Quantization is a technique that reduces memory requirements and computational overhead by converting 32-bit floating-point weights and activations into low-bit precision (Gholami et al., 2022; Kim & Kim, 2021). The commonly used uniform symmetric quantization can be expressed as follows:

$$Q(x) = \mathbf{X} \cdot s, s = \frac{2^b - 1}{\alpha - \beta}, \tag{1}$$

where $\mathbf{X}$ represents the input activation, and $s$ is the scaling factor. $\alpha$ and $\beta$ denote the maximum and minimum values, respectively. $b$ represents the number of quantization bits. Quantization is the process of mapping data from the range of 32-bit floating-point values into a lower-bit integer domain. However, when large outliers are present, the quantization step can become excessively large, leading to increased quantization error (Ki & Kim, 2022). Therefore, it is crucial to minimize accuracy degradation by handling these outliers while reducing bit precision. To address

outliers effectively, (Xiao et al., 2023) used a method for smoothing outliers in activations. (Kim et al., 2024) introduced quantization-aware distribution scaling (QADS) to handle outliers in MobileNet (Sandler et al., 2018) blocks. (Li et al., 2023) introduced a quantization approach leveraging a reparameterization-based distribution scaling technique. However, previous quantization studies do not consider the characteristics of the model after LRA, although we observed large outliers in specific channels and tokens after LRA. This means that quantizing activations with outliers may severely decrease model performance. To address these issues, we propose a novel distribution scaling method and demonstrate that per-token quantization is suitable for LRA.

## 3. Proposed Method

### 3.1. Singular Value Decomposition-based Low-Rank Approximation

First, we describe an SVD-based LRA. Specifically, the SVD for FC layers with weight matrix $w \in \mathbb{R}^{m \times n}$ is expressed as follows:

$$w = USV^T \tag{2}$$

The matrix $w$ is decomposed into $U \in \mathbb{R}^{m \times m}$, $S \in \mathbb{R}^{m \times n}$, and $V \in \mathbb{R}^{n \times n}$. $S$ is a diagonal matrix consisting of $min(m, n)$ singular values. Matrices $U, V$ represent the left and right singular vectors, respectively, and are orthonormal matrices. LRA is performed using only the top $r$ diagonal elements of the $S$ matrix. The SVD-based LRA can be expressed as follows:

$$\begin{cases} U^{'} = U_{[:,:r]} S_{[:r,:r]}^{1/2} \\ V^{'} = (S_{[:r,:r]}^{1/2} V_{[:r,:]})^T \end{cases} \tag{3}$$

where $U^{'} \in \mathbb{R}^{m \times r}, V^{'} \in \mathbb{R}^{n \times r}$ are the low-dimensional decompositions of the matrix $w$, respectively. In this case, the total number of parameters changes from $O(mn)$ to $O(r(m+n))$. Since $r < min(m, n)$, this results in a significant reduction in the number of parameters. Consequently, we obtain the $y = w^T x \approx V^{'}(U^{'T} x)$.

### 3.2. Reparameterizable Branch-based Low-Rank Approximation

In Eq. (3), the operations of the FC layer involving the approximated $V^{'}$ and $U^{'}$ can be expressed in the following reformulated form:

$$\begin{aligned} y &= V^{'}(U^{'T} \mathbf{X}) + E\mathbf{X} \\ E\mathbf{X} &= (w^T - V^{'} U^{'T}) \mathbf{X} \end{aligned} \tag{4}$$

The matrix $E$ denotes the difference between the foundation model's weight matrix and the LRA matrix. Although

directly computing the $E$ matrix preserves accuracy, it introduces additional parameters during the inference process, thereby negating the memory reduction advantages. Therefore, we design the matrix $\widetilde{E}$ as a low-rank residual branch and propose the RB-LRA method derived from this design:

$$\begin{aligned} y &\approx V^{'}(U^{'T}\mathbf{X}) + \widetilde{E}\mathbf{X} = (V^{'} + \widetilde{V})(U^{'T} + \widetilde{U}^T)\mathbf{X} \\ &where \ \widetilde{E}\mathbf{X} = (V^{'}\widetilde{U}^T + \widetilde{V}U^{'T} + \widetilde{V}\widetilde{U}^T)\mathbf{X} \end{aligned} \tag{5}$$

In Eq. (5), the structure is formulated as a residual branch (i.e., $\widetilde{V}, \widetilde{U}$) relative to the input $\mathbf{X}$, enabling its consolidation into a single branch via the application of a reparameterization technique :

$$y \approx V^{'}(U^{'T}\mathbf{X}) + \widetilde{E}\mathbf{X} = (V^{'} + \widetilde{V})(U^{'T} + \widetilde{U}^T)\mathbf{X} = A^{'}B^{'T}\mathbf{X} \tag{6}$$

The problem of finding the optimal $\widetilde{E}$ can be simplified to finding $\widetilde{V}$ and $\widetilde{U}$. We determine the optimal $\widetilde{V}$ and $\widetilde{U}$ matrices through fine-tuning.

### 3.3. Initialization Method of Reparameterizable Branch

Since the $\widetilde{V}, \widetilde{U}$ matrices are not foundation model-oriented elements, the initialization method is crucial. Therefore, we propose WR, which leverages the original weights of the foundation model that are discarded during LRA application, considering the characteristics of the FC layer operations. In Eq. (3), we remove all submatrices except for those corresponding to the top $r$ singular values and their related elements in the $U$ and $V$ matrices. The removed submatrices are expressed as follows:

$$\begin{cases} U_{del} = U_{[:,r:]} \\ S_{del} = S_{[r:,r:]} \\ V_{del} = V_{[r:,:]} \end{cases} \tag{7}$$

We obtain the matrices $U, S$, and $V$ by applying SVD. The value $r$ satisfies $r < min(m, n)$. The process of reconstructing the original FC layer using the removed submatrices and the LRA matrix is expressed as follows:

$$\begin{aligned} y^{'} &= U^T\mathbf{X} = Concat(U^{'T}\mathbf{X}, U_{del}^T\mathbf{X}) \\ y &= VS^Ty^{'} = V^{'}y^{'}_{[:r,:]} + (S_{del}V_{del})^T y^{'}_{[r:,:]} \end{aligned} \tag{8}$$

We exploit the property that the $(S_{del}V_{del})^T$ matrix can be reconstructed via addition, and utilize its value to initialize the $\widetilde{V}$ matrix. Conversely, since the $U_{del}^T$ matrix is constructed through concatenation, it is not fully reconstructed. As a result, the $\widetilde{U}$ matrix is initialized to zero and then optimized through fine-tuning. When comparing the sizes of the weight elements removed after applying the compression method, the LRA removes $O(m^2 + n^2 - r(m + n))$ weight elements. In contrast, the proposed WR method removes only $O(m(n - r))$ weight elements. As a result, the proposed WR method can significantly reduce information loss

compared to LRA. Consequently, unlike conventional LRA methods, our method can be fine-tuned to achieve optimal performance with minimal loss of weight matrix information and effectively reduce parameters during inference. In addition, a theoretical analysis of the proposed RB-LRA and WR methods is provided in Appendix A.1.

### 3.4. Block-Level Knowledge Distillation

After applying RB-LRA, we perform fine-tuning using KD to achieve high accuracy. When applying KD, selecting the appropriate teacher model and determining which features from the teacher model to distill are critical considerations. RB-LRA compresses a large-scale original model into a shallower model. In this scenario, high accuracy can be achieved by utilizing the large-scale original model as the teacher model. Distilling layer-wise feature knowledge from the teacher model can lead to overfitting during the fine-tuning process (Liu et al., 2023). Therefore, we apply encoder block-level KD, with the loss function designed as follows:

$$\mathcal{L}_{kd} = MSE(f_t^l(x_l), f_s^l(x_l)) \tag{9}$$

where $f_t^l$ represents the output of the $l$-th encoder in the teacher model. $f_s^l$ represents the output of the $l$-th encoder in the student model, and $x_l$ denotes the input to the $l$-th encoder. The final loss function considering block-level KD loss can be expressed as follows:

$$\mathcal{L} = \alpha\mathcal{L}_{ce} + \beta\mathcal{L}_{kd} \tag{10}$$

where $\mathcal{L}_{ce}$ represents the commonly used cross-entropy loss (Cox, 1958). $\alpha$ and $\beta$ denote the scale terms of the loss, respectively. For simplicity, we set both $\alpha$ and $\beta$ to 1. Finally, we use a combination of RB-LRA and block-level KD to improve accuracy. In particular, unlike previous methods, the RB-LRA method achieves higher accuracy by applying block-wise KD while minimizing the loss of information caused by applying LRA.

### 3.5. Low-Rank Approximation-Aware Quantization

We use the commonly used configuration of 8-bit per-channel and per-layer uniform quantization for each of the weights and activations, respectively, as the baseline (Lin et al., 2021). In the baseline model, we observed a drop in accuracy of 2.19% and 1.30% for Swin-T and Swin-B, respectively, compared to RB-LRA with 32-bit floating-point precision. To analyze the causes of this accuracy drop, we measured quantization error per layer. Consequently, we found that the RB-LRA layer exhibited a significant quantization error. To identify the cause, we analyzed the characteristics of the activation distribution by visualizing the distribution in the RB-LRA layer. Figure 3 shows the activation distribution of a specific RB-LRA layer in the

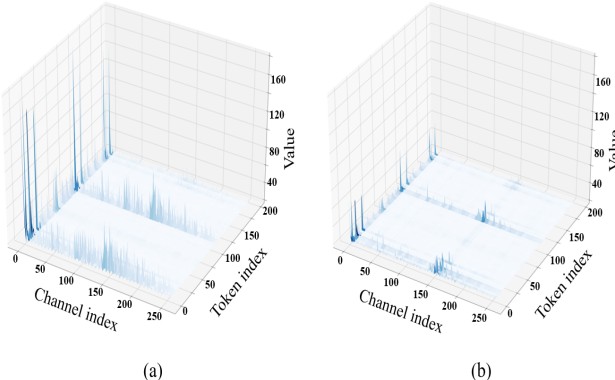

(a)        (b)

*Figure 3.* Visualization of activation distribution after applying RB-LRA to the Swin-B model. The x-axis represents the channel number, the y-axis represents the token number, and the z-axis represents the absolute value. (a) shows the output activation distribution after applying RB-LRA to the 11th encoder block. In this case, outliers become more severe in certain channels. (b) shows the output activation distribution after applying WADS to the 11th encoder block, showing a reduction in outliers compared to (a).

Swin-B model. Based on these observations, we identified two main causes of the decrease in accuracy.

First, as shown in Figure 3, we found outliers in certain channels of the RB-LRA layer activation in certain encoder blocks. Based on these observations, we propose an activation scaling method that uses a channel scaling vector to mitigate the large outliers in specific channels, as follows:

$$\mathbf{Y}_q = Q(\mathbf{X}/\alpha)Q(\alpha\mathbf{W}) \tag{11}$$

where $\alpha$ and $Q(\cdot)$ represent the channel scaling vector and the quantization function, respectively. $\mathbf{X}, \mathbf{W}$, and $\mathbf{Y}_q$ represent the input activation, weight, and output activation, respectively. The loss function to determine the optimal $\alpha$ that minimizes the quantization error is expressed as follows (Kim et al., 2024) :

$$\mathcal{L}(\alpha) = \left\|Q(\mathbf{X}/\alpha)Q(\alpha\mathbf{W}) - \mathbf{XW}\right\|^2 \tag{12}$$

where $Q(\cdot)$ represents the quantization function. $\mathbf{X}$ represents the input activation, and $\mathbf{W}$ represents the weight. By using this method, the activation quantization error can be minimized. However, when applying this method to a model with RB-LRA, the weight quantization error becomes severe in certain layers. Therefore, we propose WADS to solve this problem. First, we measure the quantization loss of all RB-LRA layer weights. The quantization loss of the weights can be measured as follows:

$$\mathcal{L}(w) = \left\|Q(\mathbf{W}) - \mathbf{W}\right\|^2 \tag{13}$$

where $\mathbf{W}$ and $Q(\cdot)$ represent the weight before quantization and quantization function, respectively. We only apply activation scaling for layers with smaller than average values of

| Model | Method | KD Method | Params(M) | GFLOPs | ACC.(%) | Diff.(%) |
|---|---|---|---|---|---|---|
| DeiT-T | Baseline | - | 5.7 | 2.2 | 72.17 | - |
| | LRA | - | 5.2 (-8.8%) | 1.8 | 68.40 | -3.77 |
| | **RB-LRA** | **-** | **5.2 (-8.8%)** | **1.8** | **70.92** | **-1.25** |
| | **RB-LRA + KD** | **Feature** | **5.2 (-8.8%)** | **1.8** | **71.70** | **-0.47** |
| DeiT-B | Baseline | - | 86.6 | 33.7 | 81.85 | - |
| | LRA | - | 44.4 (-45.7%) | 17.1 | 78.76 | -3.09 |
| | PELA (Guo et al., 2024) | Feature | 44.1 (-49.1%) | 17.0 | 81.00 | -0.85 |
| | **RB-LRA** | **-** | **44.4 (-45.7%)** | **17.1** | **79.93** | **-1.92** |
| | **RB-LRA + KD** | **Feature** | **44.4 (-45.7%)** | **17.1** | **81.12** | **-0.73** |
| Swin-T | Baseline | - | 28.3 | 8.6 | 81.37 | - |
| | LRA | - | 21.1 (-25.4%) | 6.7 | 77.30 | -4.07 |
| | **RB-LRA** | **-** | **21.1 (-25.4%)** | **6.7** | **80.27** | **-1.1** |
| | **RB-LRA + KD** | **Feature** | **21.1 (-25.4%)** | **6.7** | **80.49** | **-0.88** |
| Swin-B | Baseline | - | 88.1 | 30.3 | 83.47 | - |
| | LRA | - | 60.1 (-31.8%) | 21.1 | 81.75 | -1.72 |
| | AAFM+GFM (Yu & Wu, 2023) | Feature | 60.2 (-31.7%) | - | 82.99 | -0.48 |
| | PELA (Guo et al., 2024) | Feature | 62.2 (-29.4%) | 21.3 | 82.50 | -0.97 |
| | **RB-LRA** | **-** | **60.1 (-31.8%)** | **21.1** | **82.88** | **-0.59** |
| | **RB-LRA+KD** | **Feature** | **60.1 (-31.8%)** | **21.1** | **83.44** | **-0.03** |

*Table 1.* Comparison of Top-1 Accuracy (%) across various models with RB-LRA applied on ImageNet dataset.

$\mathcal{L}(w)$ to reduce the accuracy drop owing to weight quantization errors. Furthermore, we redesign the loss function to find the optimal channel scaling vector. The process of determining the optimal channel scaling vector $\alpha'$ is expressed as follows:

$$\alpha' = \arg\min_{\alpha} \left\{ \mathcal{L}(\alpha) + \left\| Q(\alpha \mathbf{W}) - \mathbf{W} \right\|^2 \right\} \quad (14)$$

where $L(\alpha)$ refers to Eq. (12). This prevents $\alpha$ from becoming excessively large during optimization, which may cause outliers in the weights. The optimal channel scaling vector is determined using only a subset of the training dataset, enabling optimization without requiring access to the full dataset. As a result, by utilizing an activation scaling method that considers the weights quantization errors, we achieve better performance compared to the previous activation scaling method.

Meanwhile, we also found that outliers emerged in specific tokens. If per-layer quantization is applied to activations with these outliers, the quantization error may be substantial; however, using a finer granularity than per-layer quantization can help reduce quantization errors. Therefore, we propose that per-token quantization is an appropriate method when considering token outliers and integer domain FC layer operations. Consequently, the proposed WADS method and per-token activation quantization effectively mitigate the accuracy degradation caused by outliers when applying RB-LRA.

## 4. Experimental Results

### 4.1. Comparison Results on Various Networks

**RB-LRA for Image Classification Tasks:** We evaluated the performance by applying RB-LRA to the DeiT (Touvron et al., 2021) and Swin transformer (Liu et al., 2021). Specifically, RB-LRA was applied to all the FC layers within the encoder blocks of each model. All experiments used an initial learning rate of 1e-5, the AdamW optimizer (Loshchilov & Hutter, 2017), and the cosine annealing scheduler (Loshchilov & Hutter, 2016). Experiments were conducted on a single A100 GPU within the TiMM (Wightman, 2019) environment.

To demonstrate the effectiveness of our proposed method combining RB-LRA and block-level KD (*i.e.*, RB-LRA+KD) on image classification tasks, Table 1 compares its performance with state-of-the-art (SOTA) LRA methods, including PELA (Guo et al., 2024) and AAFM + GFM (Yu & Wu, 2023). As the AAFM+GFM method does not utilize the full dataset, its performance is compared exclusively with that of the Swin-B model, for which full-dataset performance is reported in the referenced paper.

In addition, to demonstrate the mitigation of information loss in the LRA method, we compared the performance of RB-LRA with that of the conventional LRA (*i.e.*, naive SVD-based LRA) without applying the KD method. The experimental results show that applying RB-LRA and block-level KD enables the DeiT-T and DeiT-B models to reduce parameters by 8.8% and 45.7% and giga floating-point operations (GFLOPs) by 18.2% and 49.3%, with only 0.47% and 0.73% accuracy drop, respectively. For the Swin-T and Swin-B models, RB-LRA+KD achieved parameter reductions of

| Model | Method | Prec. | Size(MB) | ACC.(%) | Diff.(%) |
|-------|--------|-------|----------|---------|----------|
| DeiT-T | Baseline(RB-LRA) | FP32 | 20.8 | 71.70 | - |
| | NaivePTQ | | | 70.90 | -0.80 |
| | SmoothQuant (Xiao et al., 2023) | | | 71.43 | -0.27 |
| | Repq-ViT (Li et al., 2023) | INT8 | 5.2 | 71.38 | -0.32 |
| | QADS (Kim et al., 2024) | | | 71.40 | -0.30 |
| | **WADS** | | | **71.52** | **-0.18** |
| DeiT-B | Baseline(RB-LRA) | FP32 | 177.6 | 81.12 | - |
| | NaivePTQ | | | 79.62 | -1.50 |
| | SmoothQuant (Xiao et al., 2023) | | | 80.26 | -0.86 |
| | Repq-ViT (Li et al., 2023) | INT8 | 44.4 | 80.37 | -0.75 |
| | QADS (Kim et al., 2024) | | | 79.82 | -1.30 |
| | **WADS** | | | **80.56** | **-0.56** |
| Swin-T | Baseline(RB-LRA) | FP32 | 84.4 | 80.49 | - |
| | NaivePTQ | | | 78.30 | -2.19 |
| | SmoothQuant (Xiao et al., 2023) | | | 80.00 | -0.49 |
| | Repq-ViT (Li et al., 2023) | INT8 | 21.1 | 80.08 | -0.41 |
| | QADS (Kim et al., 2024) | | | 80.04 | -0.45 |
| | **WADS** | | | **80.20** | **-0.29** |
| Swin-B | Baseline(RB-LRA) | FP32 | 240.4 | 83.44 | - |
| | NaivePTQ | | | 82.14 | -1.30 |
| | SmoothQuant (Xiao et al., 2023) | INT8 | 60.1 | 82.76 | -0.68 |
| | QADS (Kim et al., 2024) | | | 82.37 | -1.07 |
| | **WADS** | | | **82.97** | **-0.47** |

*Table 2.* Comparison of size (MB) and accuracy between naive quantization, previous quantization methods, and proposed WADS.

25.4% and 31.8%, respectively, along with GFLOPs reductions of 22.1% and 30.4%. These reductions were achieved with minimal accuracy drops of only 0.88% and 0.03%, respectively. As a result, the proposed method demonstrated higher accuracy and greater robustness compared to conventional LRA approaches. Furthermore, we demonstrate that our method performs well on tiny networks.

**Quantization:** In this subsection, we analyze the performance of 8-bit naive post-training quantization (NaivePTQ) and the proposed WADS after applying RB-LRA. We also compare WADS with the SmoothQuant (Xiao et al., 2023), Repq-ViT (Li et al., 2023), and QADS (Kim et al., 2024) methods to demonstrate WADS effectiveness. The proposed method applies WADS to models trained using both RB-LRA and block-level KD. According to the previous analysis, per-token quantization is applied to activations, and per-channel quantization is applied to weights. We applied static quantization and performed calibration using 32 data samples to determine the optimal scaling factor. Subsequently, we conducted WADS optimization using the same dataset. This process is performed layer by layer for RB-LRA, finding the optimal channel scaling vector that minimizes quantization loss through 300 iterations of scaling vector updates. As shown in Table 2, WADS significantly mitigates accuracy drops across all ViT-based models under the same model size. In particular, DeiT-B and Swin-B achieved 0.94% and 0.83% higher accuracy, respectively, compared to NaivePTQ. In addition, our proposed method achieved the highest accuracy across all models compared to

all other methods. This indicates that WASD, which considers the weight quantization error, is the optimal method for combination with RB-LRA. Furthermore, a comprehensive analysis of the variations in quantization error due to the application of WADS is provided in Appendix A.3, while the analysis of performance variations across different quantization methods (*i.e.*, WADS, per-token) is discussed in A.4.

### 4.2. Performance Comparison with SOTA Compression Methods

We evaluate the proposed framework on the Swin-B model, comparing its performance against SOTA methods in both quantization and LRA. Since these approaches differ in their original baseline accuracies, we report accuracy degradation relative to each method's respective baseline to ensure fair comparison. As shown in Table 3, our method achieves an accuracy of 82.97%, while simultaneously applying LRA and quantization—two compression techniques that are typically challenging to integrate without significant accuracy loss. Compared to existing LRA-based methods such as PELA and AAFM+GFM, this result demonstrates a highly competitive trade-off between model size and accuracy. Furthermore, compared to existing quantization methods, our approach minimizes accuracy degradation even with a smaller model size of 6.1 MB. These results demonstrate that the proposed framework achieves competitive performance compared to both SOTA LRA methods and other representative compression methods.

| Method | LRA | Quant | Train | Model Size(MB) | Baseline Acc. (%) | Acc. (%) |
|---|---|---|---|---|---|---|
| PTQ4ViT (Yuan et al., 2022) | | ✓ | | | | 84.18 (-1.09) |
| APQ-ViT (Ding et al., 2022) | - | ✓ | - | 66.1 | 85.27 | 84.01 (-1.26) |
| QDrop (Wei et al., 2022) | | ✓ | ✓ | | | 84.33 (-0.94) |
| I&S-ViT (Zhong et al., 2023) | | ✓ | ✓ | | | 84.94 (-0.33) |
| PELA (Guo et al., 2024) | ✓ | | ✓ | 248.8 | | 82.50 (-0.97) |
| AAFM+GFM (Yu & Wu, 2023) | ✓ | - | ✓ | 240.8 | 83.47 | 82.68 (-0.79) |
| **Ours** | **✓** | **✓** | **✓** | **60.1** | | **82.97 (-0.5)** |

*Table 3.* Comparison of size (MB) and Top-1 accuracy on the ImageNet dataset between the proposed framework and SOTA compression methods using the Swin-B Model.

| Model | Method | Prec. | Size(MB) | Android(ms) | Xavier(ms) |
|---|---|---|---|---|---|
| | Baseline | FP32 | 346.4 | 275.6 | 150.7 |
| DeiT-B | RB-LRA | FP32 | 177.6 | 153.2 | 73.6 |
| | RB-LRA + WADS | INT8 | 44.4 | 86.7 | 59.4 |
| | Baseline | FP32 | 113.2 | 98.5 | 61.1 |
| Swin-T | RB-LRA | FP32 | 84.4 | 83.6 | 38.6 |
| | RB-LRA + WADS | INT8 | 21.1 | 67.3 | 27.4 |
| | Baseline | FP32 | 352.4 | 287.4 | 140.5 |
| Swin-B | RB-LRA | FP32 | 240.4 | 226.3 | 102.2 |
| | RB-LRA + WADS | INT8 | 60.1 | 155.3 | 96.2 |

*Table 4.* Latency (ms) comparison across methods for executing the DeiT and Swin Transformer models on mobile and edge devices.

## 4.3. Latency Analysis on Mobile and Edge Devices

We validate the practical efficiency of our proposed method by measuring latency on real mobile and edge devices. Specifically, latency measurements were conducted on an Android smartphone for the mobile device and on the NVIDIA Jetson Xavier platform for the edge device. In particular, latency on the Android smartphone was measured using the 3.36 GHz Cortex-X3 main core, while the TensorRT engine was employed for latency measurements on the NVIDIA Jetson Xavier platform. Table 4 shows that the model incorporating both RB-LRA and WADS achieved inference speedups ranging from $1.9\times$ to $3.2\times$ on mobile devices and from $1.5\times$ to $2.5\times$ on edge devices. These results validate that the proposed RB-LRA and WADS-based quantization method not only optimizes the trade-off between accuracy and memory usage but also delivers significant acceleration benefits, making it practical for deployment in mobile and edge device environments.

## 4.4. Ablation Studies

**Evaluation on detection and segmentation tasks:** To demonstrate the effectiveness of our proposed method, we conducted performance evaluations for detection and segmentation tasks using the MSCOCO dataset (Lin et al., 2014). Specifically, we used an ImageNet pre-trained Swin-T model as the backbone network for the Mask-RCNN model. We then applied RB-LRA to the backbone network before training. Experiments were conducted in the MMDe-

tection (Chen et al., 2019) environment. For a comprehensive analysis, we compared the performance of the proposed method with CNN-based ResNet (He et al., 2016) and ViT-based PVT (Wang et al., 2021) backbone networks. As shown in Table 5, our method achieves outstanding performance even on these downstream tasks. In the detection task, we achieved an average precision (AP) of 42.5 with a 15.1% reduction in parameters and a 6.1% reduction in GFLOPs. This represents only a 0.2 AP drop compared with the Swin-T backbone network. Additionally, our method outperforms the ResNet-50 backbone network by 2.5 AP while using 3.8 M fewer parameters. In the segmentation task, our method achieved an AP of 39.0, which is equivalent to the accuracy of the PVT-M backbone network. However, it offers a better parameter-accuracy trade-off, reducing both parameters and computational cost (GFLOPs) by 36.5% and 31.3%, respectively.

To further demonstrate the generalizability of our approach to diverse applications, we evaluate its effectiveness on the pose estimation task. In particular, we integrate RB-LRA into the ViTPose-B (Xu et al., 2022), followed by fine-tuning and evaluation on the MSCOCO Keypoint dataset. As shown in Table 6, applying RB-LRA achieves a 25.7% reduction in model size, with only a 0.9% and 0.6% drop in AP and average recall (AR), respectively. These results highlight the effectiveness of our method in preserving performance while significantly reducing model complexity, even in structurally distinct vision tasks.

| Backbone | Params(M) | GFLOPs | $AP^{box}$ | $AP^{mask}$ |
|---|---|---|---|---|
| ResNet-50 (He et al., 2016) | 44.4 | 250.2 | 40.0 | 36.1 |
| PVT-M (Wang et al., 2021) | 63.9 | 351.2 | 42.0 | 39.0 |
| Swin-T (Liu et al., 2021) | 47.8 | 256.8 | 42.7 | 39.3 |
| **Swin-T + RB-LRA** | **40.6** | **241.2** | **42.5** | **39.0** |

*Table 5.* Performance comparison of Mask-RCNN models with various backbone networks on the COCO validation dataset.

| Model | Method | Params(M) | AP | AR |
|---|---|---|---|---|
| ViTPose-B | Baseline | 89.9 | 75.9 | 81.0 |
| | **RB-LRA** | **66.8 (-25.7%)** | **75.0** | **80.4** |

*Table 6.* Performance of ViTPose with RB-LRA on the COCO Keypoint Dataset.

| Model | Init. Method | Acc.(%) | Diff.(%) |
|---|---|---|---|
| | Baseline | 81.37 | - |
| Swin-T | Random | 79.54 | -1.83 |
| | **WR** | **80.49** | **-0.88** |
| | Baseline | 83.47 | - |
| Swin-B | Random | 82.65 | -0.82 |
| | **WR** | **83.44** | **-0.03** |

*Table 7.* Comparison of Top-1 Accuracy (%) on the ImageNet dataset under varying initialization methods for the $\widetilde{V}$ branch.

| Model | Method | Params(M) | PPL | WER |
|---|---|---|---|---|
| GPT-2 Medium | Baseline | 354.8 | 18.72 | - |
| | **RB-LRA** | **249.4 (-29.7%)** | **19.51** | - |
| Conformer-L | Baseline | 116.8 | - | 5.4 |
| | **RB-LRA** | **86.2(-26.3%)** | - | **5.6** |

*Table 8.* Evaluation of PPL on the Wikitext-103 dataset for GPT-2 Medium and WER on the LibriSpeech dataset for Conformer-L, with RB-LRA applied.

the training split of the respective datasets. As shown in Table 8, despite achieving a compression ratio of 26%∼30%, the proposed method incurs less than a 1% increase in PPL and WER. Ultimately, these results demonstrate that our method extends beyond computer vision tasks, exhibiting strong generalizability and scalability across diverse modalities.

## 5. Conclusion

In this paper, we propose a compression solution aimed at enabling memory- and power-efficient deployment of ViT models in resource-constrained environments, such as mobile and edge devices. The proposed RB-LRA method introduces a reparameterizable branch specifically optimized to address errors induced by LRA. Furthermore, the WR method initializes the weights of the RB-LRA branch using the weights discarded during the LRA process, effectively minimizing information loss. Additionally, by combining RB-LRA with block-level KD, we effectively balance the trade-off between compression ratio and accuracy. Subsequently, we applied optimized quantization that considers the characteristics of LRA. We observed large outliers in certain channels and tokens of the RB-LRA output activations. Based on these observations, we proposed WADS to mitigate these outliers. Through various experiments, we demonstrated that WADS significantly reduces accuracy loss. Additionally, we found that the per-token quantization method effectively handles token outliers making it suitable for quantizing LRA layers. Consequently, our approach, which integrates RB-LRA with our proposed quantization techniques, significantly reduces model size while maintaining high accuracy compared to the baseline. As a result, this solution enables the efficient deployment of various transformer-based models in mobile and edge environments.

**Analysis of performance variations based on initialization methods:** We conduct a performance analysis of the WR initialization method. Specifically, after integrating RB-LRA into the Swin-T and Swin-B models, the $\widetilde{V}$ branch weights are initialized using either the WR method or a random initialization approach. In this process, random initialization weights $W \sim \mathcal{N}(0,1)$ are utilized. Subsequently, fine-tuning is performed using the same KD method. As shown in 7, the WR method achieves higher accuracy compared to the random initialization method. This demonstrates that fine-tuning with the WR method, which reduces information loss, is an effective approach for achieving optimal accuracy. A more detailed analysis is provided in Appendix A.2.

**Evaluation of generalization capability across diverse modalities**

To assess the cross-domain generalizability of the proposed RB-LRA method beyond computer vision tasks, we evaluate its effectiveness on both language and speech processing tasks. For language modeling, we apply RB-LRA to the GPT-2 Medium (Radford et al., 2019) and measure perplexity (PPL) on the Wikitext-103 dataset (Merity, 2016). For speech recognition, we integrate RB-LRA into the Conformer-L (Gulati et al., 2020) and evaluate word error rate (WER) on the LibriSpeech test-clean dataset (Panayotov et al., 2015). In both cases, fine-tuning is performed using

## Acknowledgements

This research was partly supported by the MSIT(Ministry of Science and ICT), Korea, under the ITRC(Information Technology Research Center) support program (IITP-2025-RS-2022-00156295) supervised by the IITP(Institute for Information & Communications Technology Planning & Evaluation) and the Technology Innovation Program(or Industrial Strategic Technology Development Program)(RS-2025-02307330, Development and Validation of On-Device AI Semiconductor for Manufacturing Automation Robots through sVLM-Based Situational Awareness) funded By the Ministry of Trade Industry & Energy(MOTIE, Korea).

## Impact Statement

This paper presents work whose goal is to advance the field of Machine Learning. There are many potential societal consequences of our work, none which we feel must be specifically highlighted here.

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

## A. Additional Analysis of the Proposed Method

### A.1. Theoretical Analysis of RB-LRA and WR

We conduct a theoretical analysis of the proposed RR-LRA and WR methods. Specifically, we begin by reconstructing the LRA matrix and subsequently evaluate the reconstruction error as follows:

$$W_r := \sum_{i=1}^{r} \sigma_i u_i v_i^T, \quad \|W - W_r\|_F^2 = \left\|\sum_{i=r+1}^{\min(m,n)} \sigma_i u_i v_i^T\right\|_F^2 = \sum_{i=r+1}^{\min(m,n)} \sigma_i^2 \tag{15}$$

Here, $W_r \in \mathbb{R}^{m \times n}$ represents the reconstructed LRA matrix. Eq. (15) follows from the Eckart–Young–Mirsky Theorem (Eckart & Young, 1936), and the formulation involving the error matrix $E$ is expressed as follows:

$$\|W - (W_r + E)\|_F^2 = \left\|\sum_{i=r+1}^{\min(m,n)} \sigma_i u_i v_i^T - E\right\|_F^2 \tag{16}$$

That is, the optimal error matrix $E$ satisfying the following condition can be obtained in a closed form:

$$\min_{E}\|W - (W_r + E)\|_F^2 \tag{17}$$

However, the error matrix $E$ directly derived from Eq. (17) has a parameter size of $\mathcal{O}(mn)$, and incorporating it into the model increases the total parameter size to $\mathcal{O}(2mn)$, thereby undermining the primary objective of applying LRA. Accordingly, we propose a reparameterizable error matrix $\tilde{E}$, formulated using a low-rank residual-branch structure to preserve the efficiency of LRA as follows:

$$\tilde{E} = \left(V\tilde{U}^T + \tilde{V}U^T + \tilde{V}\tilde{U}^T\right) \tag{18}$$

When the proposed RB-LRA and WR are applied, the error matrix $\tilde{E}$ can be formulated as follows:

$$\tilde{E} = VU' \approx \sum_{i=r+1}^{r+l} \sigma_i u_i v_i^T, \quad \text{with } l < \min(m,n) - r \tag{19}$$

Consequently, the reconstruction error resulting from the application of the proposed WR method is formulated as follows:

$$W_{\text{WR}} = W_r + \tilde{E} \approx \sum_{i=1}^{r+l} \sigma_i u_i v_i^T \tag{20}$$

Consequently, Eq. (16) leads to the following expression:

$$\|W - (W_r + \tilde{E})\|_F^2 < \|W - W_r\|_F^2 = \sum_{i=r+1}^{\min(m,n)} \sigma_i^2 \tag{21}$$

As demonstrated in Eq. (21), the proposed RB-LRA and WR methods achieve a more effective reduction in reconstruction error compared to the conventional SVD-based LRA. This implies that the proposed approach minimizes information loss, thereby playing a critical role in accelerating convergence and achieving optimal accuracy during the fine-tuning process.

### A.2. Analysis of the Impact of Branch Initialization Method on Activation

We analyze the impact of the initialization method for the $\widetilde{V}$ branch on the output activations of the FC layer. Specifically, we apply both the WR initialization and random initialization methods to the $\widetilde{V}$ branch, and visualize the output activations of the FC layer to analyze their impact on the activation patterns. Additionally, we visualize the output activations of the FC layer with the SVD-based LRA method applied, in order to compare its effects with those of the WR initialization method. Figure 4 illustrates the FC layer activations based on the initialization method of the $\widetilde{V}$ matrix. We employed the DeiT-B

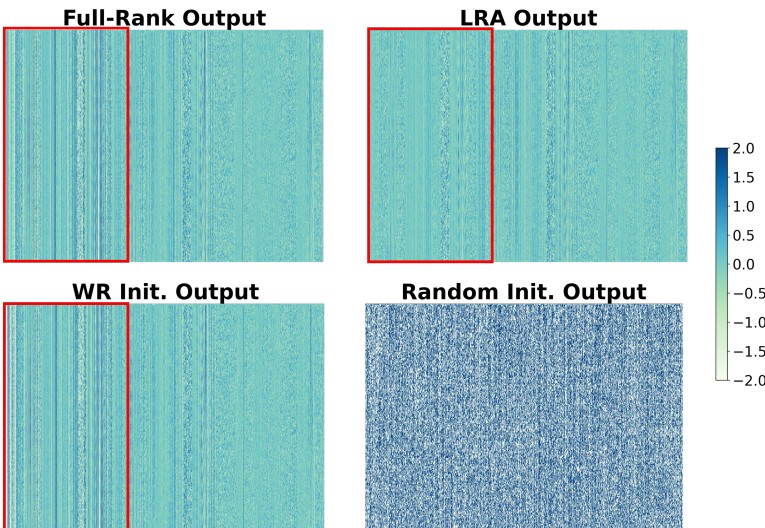

*Figure 4.* Visualization of the FC layer activations in the DeiT-B model with RB-LRA applied. The x-axis represents the dimension index, while the y-axis corresponds to the token index.

model with RB-LRA applied and conducted comparisons without fine-tuning. Furthermore, to improve the visibility of the visualizations, the data was normalized within the range of $[-2, 2]$. First, by comparing the activation visualization results of the WR method and the random initialization method, we observe that the WR method generates output similar to that of the full-rank model. In contrast, the use of the random initialization method leads to noise, causing the activation characteristics of the full-rank model to be lost. Ultimately, the presence of noise-like activations in the FC layer suggests that they hinder the attainment of optimal accuracy during the fine-tuning process. Additionally, when compared to the SVD-based LRA method, the WR method is found to better preserve the activation characteristics of the full-rank model. This can be attributed to the effectiveness of the WR method in reducing weight information loss. Consequently, the use of the WR initialization method promotes better retention of the full-rank model characteristics, ultimately contributing to the attainment of optimal accuracy.

### A.3. Impact of WADS on Activation Quantization Error

To assess the effect of WADS on reducing activation quantization error, we perform visualization and analysis of activation quantization errors in both the DeiT and Swin transformer models. Figure 5 shows the activation quantization error for the top 20 layers exhibiting the highest activation quantization errors in the DeiT and Swin Transformer models, as visualized after normalization. As shown in Figure 5, the application of WADS leads to a reduction in activation quantization error across all layers. These visualizations and analyses collectively demonstrate that the WADS method effectively mitigates outliers and minimizes quantization error.

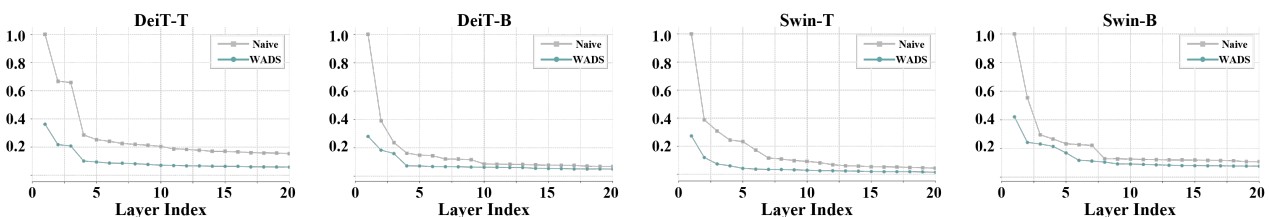

*Figure 5.* Analysis of Activation Quantization Error Changes with WADS Application. The x-axis represents the layer number, while the y-axis represents quantization error. The visualization compares the error for the top-20 layers with the highest quantization error.

## A.4. Analysis of the performance impact of each proposed quantization method

To analyze the impact of our proposed LRA-aware quantization methods (*i.e.,* WADS and per-token quantization), we performed a step-by-step classification performance evaluation on the ImageNet dataset. As shown in Table 9, applying per-token quantization to the DeiT-B and Swin-B models improved accuracy by 0.52% and 0.51%, respectively, compared to the NaivePTQ method. These improvements are attributed to the quantization process, which effectively addresses the impact of outliers on specific tokens after applying RB-LRA. Additionally, the WADS resulted in accuracy improvements of 0.42% and 0.32%, respectively. This improvement is due to WADS effectively mitigating outliers in activation-specific channels while minimizing weight quantization error. These results demonstrate that both the per-token quantization method and the WADS are effective when combined with LRA and RB-LRA, significantly improving classification performance.

| Model | Method | Per-Token | WADS | ACC.(%) |
|---|---|---|---|---|
| | NaivePTQ | X | X | 79.62 |
| DeiT-B | LRA-aware | ✓ | X | 80.14 |
| | LRA-aware | ✓ | ✓ | 80.56 |
| | NaivePTQ | X | X | 82.14 |
| Swin-B | LRA-aware | ✓ | X | 82.65 |
| | LRA-aware | ✓ | ✓ | 82.97 |

Table 9. Performance comparison of different quantization methods at each step for the RB-LRA models.

## A.5. Comparative Analysis with Existing SOTA Quantization Methods

**Accuracy Comparison with Various SOTA PTQ Methods:** We evaluate the performance of the proposed compression platform, which combines RB-LRA and WADS, in comparison to existing SOTA quantization methods. Table 10 presents a comparative analysis of the performance of various SOTA quantization methods and the proposed platform, using the FP32 DeiT-B model without LRA and quantization as the baseline. Consequently, the proposed method, which combines RB-LRA and WADS, exhibits the best performance in balancing model size and accuracy. For instance, the existing state-of-the-art method, IGQ-ViT, successfully reduced the size of the DeiT-B model to 43.3MB with 4-bit quantization, but incurred a relatively significant accuracy drop of -2.62%. In contrast, our method compressed the model to a comparable size of 44.4MB, while incurring only a minimal accuracy decrease of -1.29%. This demonstrates that the integration of LRA and quantization is highly effective in achieving model compression while maintaining accuracy. Moreover, it underscores that the proposed compression platform provides a practical and efficient alternative to existing methods.

| Method | Prec. | Size(MB) | ACC.(%) | Diff.(%) |
|---|---|---|---|---|
| BaseLine | FP32 | 346.4 | 81.85 | - |
| FQ-ViT (Lin et al., 2021) | | | 64.39 | -17.46 |
| APQ-ViT (Ding et al., 2022) | | | 67.48 | -14.37 |
| RepQ-ViT (Li et al., 2023) | INT4 | 43.3 | 75.61 | -6.24 |
| AdaLog (Wu et al., 2025) | | | 78.03 | -3.82 |
| ADFQ-ViT (Jiang et al., 2024) | | | 78.75 | -3.10 |
| IGQ-ViT (Moon et al., 2024) | | | 79.23 | -2.62 |
| **RB-LRA** | **FP32** | **177.6** | **81.12** | **-0.73** |
| **RB-LRA+WADS** | **INT8** | **44.4** | **80.56** | **-1.29** |

Table 10. Comparison of Top-1 Accuracy (%) on the ImageNet dataset across various existing SOTA quantization methods and the proposed method, evaluated on the DeiT-B.

**Comparison of Training Cost, Accuracy, and Latency Trade-offs Between SOTA PTQ Methods and the Proposed Framework:** We further compare the trade-offs between training cost, accuracy, and latency of the proposed framework against existing SOTA PTQ methods. Table 11 presents the performance of our method, FQ-ViT (Lin et al., 2021), RepQ-ViT (Li et al., 2023), and AdaLog (Wu et al., 2025) on the DeiT-B model. Latency is measured on the NVIDIA Jetson Xavier platform, while GPU time denotes the optimization time evaluated on a single A100 GPU. In terms of accuracy, the proposed method achieves the highest Top-1 accuracy of 80.56% Although it incurs a training time of 5 hours and 56 minutes, this cost is justified by the substantial accuracy improvement over existing PTQ methods. Next, in terms of

latency, existing methods fail to achieve meaningful acceleration despite applying 4-bit quantization. This is primarily because support for true 4-bit integer operations remains limited across current edge devices. In most commercial edge hardware, even when 4-bit quantization is applied at the software level, the actual computations are executed using 8-bit integer operations. Consequently, it is difficult to realize the theoretical speedup benefits of 4-bit quantization in practical deployment scenarios. In contrast, the proposed method integrates 8-bit quantization with LRA, effectively reducing both computational complexity and model size. As a result, it achieves the shortest inference latency of 59.4ms. This demonstrates that combining widely supported 8-bit integer operations with structural compression methods such as LRA can yield significant optimization benefits. These results suggest that the proposed method offers excellent practicality and computational efficiency, particularly in realistic edge deployment scenarios.

| Method | Prec. | Latency (ms) | Model Size (MB) | GPU Time | Acc.(%) |
|---|---|---|---|---|---|
| FQ-ViT (Lin et al., 2021) | | 110.9 | | 77 s | 64.39 |
| RepQ-ViT (Li et al., 2023) | INT4 | 96.8 | 43.3 | 247 s | 75.61 |
| AdaLog (Wu et al., 2025) | | 113.8 | | 2h 47m | 78.03 |
| **Ours** | **INT8** | **59.4** | **44.4** | **5h 56m** | **80.56** |

*Table 11.* Comparison of training cost, latency, and accuracy between SOTA PTQ methods and the proposed framework on the DeiT-B.

### A.6. Differences with Low-Rank Adapter-Based Parameter Efficient Fine-Tuning

Recent research has focused extensively on adapter-based methods for parameter-efficient fine-tuning (PEFT) of large Transformer models. Notably, LoRA (Hu et al., 2021) utilizes low-rank adapters to update only a limited subset of parameters, thereby significantly reducing GPU memory consumption while preserving the accuracy of full fine-tuning. QLoRA (Dettmers et al., 2024) combines LoRA with the quantization of foundation models, effectively reducing GPU memory consumption during the fine-tuning process. However, while these methods focus on reducing GPU memory usage during the fine-tuning process, they do not address the reduction of parameters in the foundation model during inference. In contrast, our approach applies LRA to the foundation model for efficient inference on mobile and edge devices, concurrently utilizing quantization to effectively reduce both the number of parameters and memory usage.

