# OpenReview forum: "LRA-QViT: Integrating Low-Rank Approximation and Quantization for Robust and Efficient Vision Transformers"
_ICML.cc/2025/Conference — ICML 2025 poster_

### Official Review · Reviewer_MFof · 2025-02-24

**Overall Recommendation:** 3

**Summary:**

This paper presents LRA-QViT, a novel framework integrating low-rank approximation (LRA) and quantization to improve the efficiency and robustness of Vision Transformers (ViTs), particularly for deployment in resource-constrained environments such as edge and mobile devices. The authors introduce Reparameterizable Branch-based Low-Rank Approximation (RB-LRA), which mitigates information loss from LRA via weight reconstruction. Additionally, they propose LRA-aware quantization, which addresses outliers induced by LRA using Weight-Aware Distribution Scaling (WADS) and per-token quantization. The method is validated through extensive experiments on ImageNet, demonstrating superior efficiency-accuracy trade-offs compared to existing compression and quantization baselines.

**Claims And Evidence:**

The claims in the paper are well-supported by experimental results:
- RB-LRA and RB-LRA + KD improve accuracy over naive LRA by introducing a residual branch and fine-tuning strategy, as shown in Table 1.
- WADS outperforms naive post-training quantization (PTQ), SmoothQuant, and QADS on RB-LRA fine-tuned models.
- The proposed method enhances practical deployment: The combination of RB-LRA and WADS achieves 1.9×–3.2× inference speedups on mobile devices and 1.5×–2.5× speedups on edge devices, all while maintaining accuracy.

**Essential References Not Discussed:**

The paper adequately discusses most essential references.

**Experimental Designs Or Analyses:**

The experimental design follows a conventional setting and includes:
- Several reasonable baselines (LoRA for fine-tuning and QADS for post-training quantization).
- Standard datasets (ImageNet and MS-COCO for downstream tasks).

One minor concern is whether it is fair to compare WADS with post-training quantization baselines. Based on the description in the paper, WADS appears to resemble quantization-aware training (QAT) rather than pure post-training quantization (PTQ).

**Methods And Evaluation Criteria:**

The methods proposed are well-justified for the problem of efficient ViT fine-tuning and compression. The evaluation is rigorous and includes:
- ImageNet classification results across DeiT and Swin Transformer models.
- Comparison with prior LRA methods.
- Comparison with quantization methods.
- Ablation studies on object detection and initialization methods.
- Latency analysis on mobile and edge devices, demonstrating real-world applicability.

**Other Comments Or Suggestions:**

N/A

**Other Strengths And Weaknesses:**

I greatly appreciate the authors' contributions to the LoRA + PTQ framework. The presentation and organization of the paper are well-structured. The experiments effectively demonstrate the impact of each component in the framework step by step:
- Comparing RB-LRA and RB-LRA + KD with LoRA.
- Comparing WADS with PTQ baselines.
- Demonstrating the memory and latency benefits of the framework on both Android and edge GPU platforms.

One minor concern is whether it is fair to compare WADS with post-training quantization baselines. Based on the description in the paper, WADS appears more similar to quantization-aware training (QAT) rather than pure post-training quantization (PTQ).

**Questions For Authors:**

- It would be good to clarify how much extra time and GPU resources the scaling optimization requires.
- How does WADS compare to alternative quantization-aware training (QAT) methods?

**Relation To Broader Scientific Literature:**

The paper builds on existing work in LRA and quantization and appropriately cites key references, such as:
- LRA techniques in ViTs (e.g., SVD-based decomposition, previous PELA methods).
- Quantization methods (e.g., SmoothQuant, RepQ-ViT, QADS).
- Knowledge distillation for model compression.

The integration of LRA and quantization into a unified framework is the core contribution of the paper.

**Theoretical Claims:**

There are no major theoretical claims requiring proof verification.

---

> ### Author Rebuttal · Authors · 2025-04-01
>
> We thank the reviewers for their valuable feedback and provide the following responses.
> # A1) Difference from QAT
> >- Our proposed WADS includes an optimization process distinct from existing PTQ methods, yet it remains fundamentally different from QAT.
> >- As shown in the right part of Figure 1, WADS first measures the layer-wise weight quantization sensitivity of the RB-LRA-applied model (Eq. 13).
> > - Then, layers with sensitivity below a certain threshold are selected, and an optimization is performed to find a scaling vector $\alpha$ that mitigates activation outliers (Eq. 14).
> >- Notably, the distinction between WADS and conventional QAT lies in the following aspect:
>
>  ---
> >**1. The method uses only 32 calibration images rather than the full training dataset, consistent with the standard static quantization flow adopted in PTQ methods.**
> >**2. WADS does not require any quantization-aware weight updates over the entire model; it only involves computing MSE-based quantization error and optimizing the scaling vector $\alpha$.**
> >**3. As the only optimization target is $\alpha$ the cost is substantially lower than QAT, which involves full model retraining.**
> ---
> >- We measured the WADS application time on an A100 GPU.
> >- Table I presents the optimization time and additional GPU memory required to optimize the scaling vector when applying WADS.
> >- As a result, the proposed WADS is clearly distinguished from QAT, which requires full training. We kindly refer the reader to Table D in our response to reviewer vjxZ for details on the full-training burden.
> >- Furthermore, the additional GPU memory overhead remains within 250 MB.
> >- As the WADS optimization is performed on high-capacity GPU hardware (e.g., 80GB A100), we believe that the additional 250MB memory usage does not present a significant bottleneck.
> >- Moreover, when deploying to edge or mobile devices, the scaling vector is fixed, eliminating the optimization burden. Therefore, this overhead does not pose a significant limitation in the context of our research objective.
> >- Accordingly, the comparison between WADS and PTQ methods presented in this paper is justified, and WADS is reaffirmed as a practical and effective alternative within PTQ settings.
>
> **Table I**
> >|Model|Method|GPU Time (s)|Additional GPU Memory (MB)|
> |-|:-:|:-:|:-:|
> |DeiT-B|WADS|160.6|192.13|
> |DeiT-T||121.7|27.24|
> |Swin-B||392.6|258.33|
> |Swin-T||131.0|88.73|
>
> # **A2) Comparison with QAT**
> >- Additionally, we analyze the trade-off between model size and accuracy of QAT methods and our proposed method.
> >-  As shown in Table J, our method effectively improves the trade-off between model size and accuracy.
> >- Moreover, from the perspective of training cost–accuracy trade-off, our method requires significantly less computational burden compared to full training, further demonstrating its superiority.
>
> **Table J**
> > |Model| Method | Prec. | Model Size (MB) | Acc.(%) |
> > |-|-|-|:-:|:-:|
> > |DeiT-B|Quantformer [1]|INT4|43.3|79.70|
> > ||I&S-ViT [2]|INT4|43.3|79.97|
> > ||Ours|INT8|44.4|80.56|
>
> # **A3) Comparison with Other Compression Methods**
>
> >- Additionally, we compare the performance of our framework with various compression methods in Table K to demonstrate its superiority.
> >- We compare performance on the Swin-B model using the INT6 quantization method to ensure a fair comparison at a similar model size.
> In addition, since baseline performance varies across methods, we compare them in terms of accuracy drop.
> >- On the Swin-B model, our method demonstrates a superior trade-off between model size and accuracy drop compared to all other INT6 PTQ methods.
> >- Although I&S-ViT exhibits a slightly smaller accuracy drop, our method achieves a more balanced and efficient trade-off when considering both model size and accuracy.
> >- Moreover, unlike conventional quantization methods, the RB-LRA shows broad applicability not only to vision tasks but also to various modality tasks, as noted in the responses from reviewers BTg6 and vjxZ. Therefore, we believe it is practically advantageous.
>
> **Table K**
> >|Model|Method|LRA|Quant|Train|Prec.|Model Size(MB)|Baseline Acc.(%)|Acc.(%)|Acc. Drop(%)|
> |-|-|:-:|:-:|:-:|:-:|:-:|:-:|:-:|:-:|
> |Swin-B|PTQ4ViT||$\checkmark$||INT6|66.1|85.27|84.18| -1.09 |
> ||APQ-ViT||$\checkmark$||INT6||| 84.01 | -1.26 |
> ||QDrop [3]||$\checkmark$|$\checkmark$|INT6|||84.33|-0.94|
> ||I&S-ViT||$\checkmark$|$\checkmark$|INT6|||84.94|-0.33|
> ||PELA|$\checkmark$||$\checkmark$| FP32|248.8|83.47|82.50|-0.97|
> ||AAFM|$\checkmark$||$\checkmark$|FP32|240.8||82.68|-0.79|
> ||Ours|$\checkmark$|$\checkmark$|$\checkmark$|INT8|60.1||82.97|-0.5|
>
> # **Reference**
> >[1] Quantformer: Learning extremely low-precision vision transformers. TPAMI'22
> [2] I&S-vit: An inclusive & stable method for pushing the limit of post-training vits quantization. arXiv’23
> [3] QDrop: Randomly Dropping Quantization for Extremely Low-bit Post-Training Quantization. ICLR'22

---

> > ### Comment · Reviewer_MFof · 2025-04-05
> >
> > I would like to thank the authors for providing detailed responses to my concerns. I maintain my positive view of this paper and will therefore keep my current rating.

---

> > > ### Author Response · Authors · 2025-04-05
> > >
> > > We sincerely appreciate your thoughtful engagement with our work and your continued positive assessment.
> > > Your acknowledgment of our rebuttal and supportive stance are deeply encouraging.
> > > Thank you once again for your time and for the constructive feedback throughout the review process.
> > > Best regards

---

### Official Review · Reviewer_xvP8 · 2025-03-13

**Overall Recommendation:** 2

**Summary:**

This paper introduces a novel framework that integrates reparameterizable branch-based Low-Rank Approximation (RB-LRA) with Knowledge Distillation (KD) to reduce the number of parameters and inference computational complexity. Additionally, the authors propose an LRA-aware post-training quantization method to enhance the performance of the model after quantization. The experimental results on the image classification task demonstrate that the proposed method achieves promising performance in both full-precision and 8-bit quantized models.

## update after rebuttal

Thank you for the authors' rebuttal.
The additional experimental results have demonstrated the promising performance of the proposed method.

However, regarding points A1) and A2), my intention was to highlight that, as shown in Table 7, the authors' method applies 8-bit quantization to the model after RB-LRA compression, while the comparison methods quantize the ViT to 4 bits. There is no direct comparison or analysis of the computational complexity between 8-bit RB-LRA and 4-bit ViT in this context. If the authors' currently available hardware does not support 4-bit quantization, a theoretical analysis of computational costs could still be conducted to evaluate the potential practical efficiency of this technique. This is particularly relevant given that 4-bit quantization is increasingly feasible for deployment on edge devices in many resource-constrained scenarios.

**Claims And Evidence:**

Some claims are not well-supported in the current version:

1. Table 7 demonstrates that the proposed method, when integrated with knowledge distillation (KD) and low-rank adaptation (LRA)-aware INT8 quantization, achieves model sizes comparable to those of INT4 quantized ViTs. However, the paper lacks an analysis of the actual computational complexity and inference speed of these models. Specifically, it would be beneficial to evaluate the real-world inference speeds of the proposed method compared to the baseline methods in Table 7 under identical hardware and model size conditions. This information is crucial for assessing the practical applicability of the proposed framework, as retraining such a model incurs significant computational costs. A well-balanced trade-off analysis between training overhead and inference efficiency would strengthen the paper’s practical contributions.
2. While the authors highlight the strong performance of their method when combined with INT8 quantization—surpassing some INT4 post-training quantization (PTQ) approaches, as shown in Table 7—there is no direct comparison between their proposed quantization method and these state-of-the-art PTQ methods. For instance, applying AdaLog [2] to the proposed full-precision model and then comparing its performance under INT8 quantization in Table 2 would provide a more comprehensive understanding of the relative strengths and limitations of the proposed approach.
3. The evaluation of the proposed method is primarily focused on image classification. A broader assessment across other tasks, such as semantic segmentation and object detection, would further substantiate its effectiveness. For example, baseline LRA methods like PELA [1] have demonstrated competitive results in these tasks, and comparing the proposed framework against them in such scenarios would provide valuable insights into its generalizability.

References
[1] Guo et al. PELA: Learning Parameter-Efficient Models with Low-Rank Approximation . In CVPR 2024.
[2] Wu et al. AdaLog: Post-Training Quantization for Vision Transformers with Adaptive Logarithm Quantizer . In ECCV 2024.

**Essential References Not Discussed:**

NA

**Experimental Designs Or Analyses:**

Some of them are not very conviencing. Please see my comments in "Methods And Evaluation Criteria"

**Methods And Evaluation Criteria:**

Some of them are not very satisfactory in current version:

1. While the authors emphasize the robust performance of their method when integrated with INT8 quantization—outperforming certain INT4 post-training quantization (PTQ) techniques, as demonstrated in Table 7—there is a notable absence of direct comparisons between their proposed quantization method and these state-of-the-art PTQ approaches.

2.The evaluation of the proposed method primarily centers on image classification. Expanding the assessment to include additional tasks, such as semantic segmentation and object detection, would further validate its effectiveness. Notably, baseline LRA methods like PELA [1] have achieved competitive performance in these domains. A comparative analysis of the proposed framework against such methods in these contexts would offer valuable insights into its generalizability.

[1] Guo et al. PELA: Learning Parameter-Efficient Models with Low-Rank Approximation . In CVPR 2024.

**Other Comments Or Suggestions:**

NA

**Other Strengths And Weaknesses:**

The proposed method shows promising performance on image classification tasks while significantly reducing inference complexity, which is a valuable contribution to ViT compression.

**Questions For Authors:**

NA

**Relation To Broader Scientific Literature:**

The paper extends prior work on low-rank approximation (LRA), knowledge distillation, and quantization in vision transformers (ViTs) by addressing key limitations in these areas. While existing LRA methods (e.g., PELA) reduce parameters through matrix decomposition, they often suffer from accuracy loss. The proposed reparameterizable branch-based LRA (RB-LRA) mitigates this by introducing weight reconstruction. Additionally, knowledge distillation, widely used in ViT compression (e.g., DeiT), is integrated to further enhance accuracy. Unlike prior LRA methods that overlook quantization, the paper introduces an LRA-aware quantization strategy to handle large outliers caused by decomposition. Empirical results on ImageNet demonstrate superior efficiency-accuracy trade-offs, making the framework highly relevant for real-world deployment.

**Theoretical Claims:**

The paper does not include any proofs or theoretical claims.

---

> ### Author Rebuttal · Authors · 2025-03-31
>
> We thank the reviewers for their valuable feedback. Refreshing the page (F5) helps generate equations properly!
>
> # A1) Computational Complexity Analysis
> >- We analyze the computational complexity (i.e., FLOPs) and inference speed improvements.
> >- Furthermore, Table 3 demonstrates the actual acceleration achieved on mobile and edge devices.
> >- Our efficiency gains stem from:
> > ### 1. RB-LRA reduces computational complexity
> >- As $\text{FLOPs} \propto \text{parameter size}$, complexity is reduced by a factor of $\frac{r(m+n)}{mn}$, which is $\ll 1$.
> >- Our method effectively reduces  FLOPs by decreasing the number of parameters, as detailed in Table 1.
> >### 2. INT8 operation compatibility enabled by quantization
> >- Specifically, when the applied quantization method is implemented using real-quantization (i.e., actual low-bit integer arithmetic rather than simulated quantization), it can be formulated as follows:
> > $$y_{[t, o]} = \sum_{t=0}^{T} \sum_{o=0}^{OC} \frac{1}{S_W[o] \cdot S_x[t]} \left( \sum_{i=0}^{IC} W_q[i, o]x_q[t, i] \right)$$
> >- $S_W[o]$ and $S_x[t]$ denote the weight scale factor and the activation scale factor, respectively
> >- With compiler support, linear layers that utilize this feature can be converted into integer operations on edge devices, thereby accelerating inference.
> # A2) Comparison of Inference Speed
> >- We measured the inference speed of the 4-bit PTQ methods on a real edge device (NVIDIA Xavier) and the results can be found in Table F.
> >- In Table F, latency refers to the inference latency on edge devices, while GPU time indicates the optimization time of RB-LRA + WADS and each quantization method in a GPU environment.
> >- As a result, the INT4 PTQ methods did not yield meaningful inference speed-up.
> >- This is because edge devices, such as the Jetson Xavier series, provide hardware acceleration up to INT8 precision.
> >- That is, even with INT4 quantization in software, edge devices perform operations similar to INT8 in practice.
> >- To the best of our knowledge, INT4 is unsupported on edge devices and only partially supported on high-end GPUs (e.g., H100).
> >- Ultimately, the proposed framework enables practical INT8 acceleration on commercial edge devices while achieving INT4-level model compression, offering greater deployment benefits.
> # A3) Training Cost, Accuracy Trade-off
> >- Table F shows trade-offs among FT cost, accuracy, and speed.
> >- The proposed method incurs the highest training cost compared to existing methods.
> However, it demonstrates a corresponding improvement in accuracy that justifies the cost.
> >- Additionally, our method achieved the highest accuracy and inference speed on real-world devices.
> >- In edge/mobile scenarios, training is typically performed on GPU servers, while inference is conducted on resource-constrained devices.
> Therefore, the trade-off between accuracy, latency, and model size becomes particularly critical in such environments.
> >- Notably, our proposed method requires only about 6 hours, which is significantly lower compared to the full baseline training cost.
> These results demonstrate that our method performs effectively in real-world deployment scenarios.
>
> **Table F**
> >|Model|Method|Prec.|Latency (ms)|Model Size (MB)|GPU Time|Acc.(%)|
> |-|-|-|:-:|:-:|:-:|:-:|
> | DeiT-B|FQ-ViT|INT4|110.9|43.3| **77 s** |64.39|
> ||RepQ-ViT|INT4|96.8|43.3|247 s|75.61|
> ||AdaLog|INT4|113.8|43.3|2h 47m|78.03|
> ||Ours|INT8|**59.4**|44.4|5h 56m|**80.56**|
> # A4) Compatibility between RB-LRA and SOTA PTQ Methods
> >- First, Table 2 already presents performance comparisons of the proposed RB-LRA model combined with QADS, SmoothQuant, RepQ-ViT, and WADS.
> >- We evaluate INT8 quantization performance by applying methods listed in Table 7, such as AdaLog, to the RB-LRA model.
> >- Since the code for APQ-ViT, ADFQ-ViT, and IGQ-ViT is not publicly available, we apply AdaLog and FQ-ViT to RB-LRA  instead.
> >- As shown in Table G, WADS shows better compatibility with RB-LRA than existing PTQ methods.
>
> **Table G**
> > |Model|Prec.|Method|Acc.(%)|
> |-|-|-|-|
> |DeiT-B|FP32|Baseline(RB-LRA)|81.12|
> ||INT8|AdaLog|79.27|
> |||FQ-ViT|77.70|
> |||WADS|**80.56**|
> # A5) Object Detection / Instance Segmentation Task
> >- We conduct experiments on object detection and instance segmentation, with results shown in Table 4.
> >- While a direct comparison with methods like PELA is difficult due to unavailable code and differing baselines (e.g., Swin-B vs. our Swin-T), we instead use accuracy drop as an indirect metric.
> >- As shown in Table H, our method shows comparable degradation, indicating that RB-LRA effectively maintains performance even with smaller models, making it competitive for mobile and edge deployment.
>
> **Table H**
> > |Method|Backbone|Model|Baseline$AP^{box}$|Baseline$AP^{mask}$|$AP^{box}$|$AP^{mask}$|
> |-|-|-|:-:|:-:|:-:|:-:|
> |AAFM|Swin-B|Cascade Mask R-CNN|52.0|45.0|51.9|44.7|
> |PELA|Swin-B|Cascade Mask R-CNN|50.1|-|49.0|-|
> |RB-LRA|Swin-T|Mask R-CNN|42.7|39.3|42.5|39.0|

---

### Official Review · Reviewer_vjxZ · 2025-03-14

**Overall Recommendation:** 3

**Summary:**

This paper proposes RB-LRA, a low-rank approximation scheme integrated with quantization to reduce the number of parameters in vision transformers (ViTs) and mitigate inference delay. To minimize approximation errors introduced by singular value decomposition (SVD), RB-LRA employs block-level knowledge distillation (KD) to fine-tune the approximated matrices. Additionally, to integrate quantization with minimal performance degradation, RB-LRA leverages weight-aware distribution scaling (WADS) to suppress weight outliers. Experimental results on ImageNet with pretrained ViTs demonstrate that RB-LRA effectively reduces the number of parameters while maintaining high accuracy.

**Claims And Evidence:**

1. The proposed RB-LRA significantly reduces LRA-induced errors by fine-tuning the approximated matrices using knowledge distillation. The accuracy drops on ImageNet for various ViT models (DeiT-T, DeiT-B, Swin-T, Swin-B) are -0.47%, -0.73%, -0.88%, and -0.03%, respectively.

2. The proposed RB-LRA effectively reduces quantization error through weight-aware distribution scaling (WADS), as validated on ImageNet across several ViT architectures.

3. The proposed RB-LRA significantly reduces inference latency by utilizing quantized approximated parameters. Experiments on Android and Xavier platforms demonstrate the effectiveness of RB-LRA.

**Essential References Not Discussed:**

This paper has cited most of the relevant literature.

**Experimental Designs Or Analyses:**

1. The experiments are comprehensive across several model architectures, bitwidth and vision tasks (classification, object detection). However, there is no analyses on the additional computation for fine-tuning the approximated matrices and optimizing the scaling factor in Eq. 14. It is more fair to compare both computation overhead and the accuracy improvement.

**Methods And Evaluation Criteria:**

1. The primary evaluation criteria include image classification accuracy on ImageNet and average precision (AP) on the COCO dataset for object detection. Model size is also considered, but it remains the same across all schemes using low-rank approximation.

2. The latency time criteria can validate the effectness of RB-LRA to accelerate inference.

**Other Comments Or Suggestions:**

1. Low-rank approximation can also be applied to language models. Evaluating the effectiveness of RB-LRA on LLMs would be valuable, as numerous text-based benchmark tasks are available.

2. Please indicate every notation after using it in the formula. For example, the $\tilde{V}$ and $\tilde{U}$  in Eq.5. You should explain what is the meaning of these two notations and why you set up in this way.

**Other Strengths And Weaknesses:**

1. The novelty is limited, as no new techniques are proposed in this work.

2. The proposed RB-LRA needs extra computation for fine-tuning with data, which restricts the application of this technique.

**Questions For Authors:**

How do you select the rank $r$? Are you selecting all the positive singular values?

**Relation To Broader Scientific Literature:**

The contribution is relatively incremental, as this paper primarily integrates widely-used techniques such as knowledge distillation and activation-aware quantization. The novelty of the work is quite limited.

**Theoretical Claims:**

There is no theoretical contribution in this paper.

---

> ### Author Rebuttal · Authors · 2025-03-31
>
> We thank the reviewers for their valuable feedback and provide the following responses.
> # A1) Additional Computation Analysis
> >- RB-LRA and WADS require one-time fine-tuning (FT) and calibration only during pre-deployment, without inference overhead.
> >- As shown in Table D, we measured the FT time of RB-LRA and compared it with the baseline training time.
> The optimization time for WADS is provided in Table I in our response to reviewer MFof.
> >- FT time and baseline training time were measured on a single A100 GPU (batch size 64, 300 epochs for baseline)
> >- RB-LRA converges within 100 epochs, requiring only 15% ~ 21% of baseline training time.
> >- Moreover, all FT and calibration were conducted on public datasets in server-like settings, aligned with common industrial workflows
> >- Overall, our method adds only minor training overhead while achieving both high accuracy and clear gains in post-deployment efficiency, which is central to our contribution.
> >- Moreover, we analyze the trade-off between FT overhead, inference speed, and accuracy improvement.
> >- The results can be found in Table F in our response to reviewer xvP8. We encourage the reviewer to refer to it
>
> **Table D**
> > |Model|Method|GPU Time|GPU Memory (GB)|
> > |-|-|-|-|
> > |DeiT-T|Baseline|20h|5.9|
> > ||RB-LRA|4h 21m|7.2|
> > |Swin-T|Baseline|35h|16.0|
> > || RB-LRA|5h 19m|19.2|
>
> # A2) Clarifying Novelty: Beyond Simple Integration
> > Our contribution lies not in simply combining existing methods, but in the distinct novelty of each component and their cohesive integration into a unified framework.
> > 1. **RB-LRA + WR**
> >- SVD-based LRA suffers from accuracy loss due to discarded components.
> >- We introduce a reparameterizable residual branch that guides FT toward optimal accuracy.
> >- During inference, it merges into a unified form, adding no extra parameters or computations.
> >- This design helps preserve accuracy while reducing memory, suitable for edge deployment.
> > 2. **WADS**
> >- While activation-aware quantization methods exist, we are the first to empirically analyze the emergence of channel- and token-level outliers following LRA.
> >- We propose a weight-aware scaling method guided by quantization loss to effectively suppress these outliers.
> > 3. **Originality in Integrated Design**
> >- Prior works typically address LRA or quantization separately.
> >- We instead propose a unified, deployment-oriented framework that systematically combines novel components such as RB-LRA and WADS.
> # A3) Language Application
> >-  Our method specifically targets the compression of linear layers, and we believe it can be broadly applied across various domains that utilize transformer-based models.
> >- In response to the reviewer’s comment, we applied RB-LRA to GPT-2 Medium on WikiText-103 using pre-trained weights from Huggingface.
> >- To further demonstrate generalizability, we applied RB-LRA to the Conformer [1] model for automatic speech recognition (ASR).
> >- For the ASR task, we used a baseline model trained from scratch on the LibriSpeech 960h dataset using our own training pipeline
> >- As shown in Table E, despite compressing model parameters by 25–30%, our method resulted in only marginal accuracy drops across both NLP and ASR tasks These results demonstrate the strong extensibility of RB-LRA to various transformer-based models and tasks.
> >- We'll include these results in the camera-ready version to demonstrate our method's generality and competitiveness.
>
> **Table E**
> > |Model|Dataset|Method|Params|Perplexity (↓)|
> > |-|-|-|:-:|:-:|
> > |GPT2|Wikitext-103|Baseline|354.8|18.72|
> > |||RB-LRA|249.4|19.51|
>
> > |Model|Dataset|Method|Params|Test-Clean WER (↓)|
> > |-|-|-|-|-|
> > |Conformer|Librispeech|Baseline|116.9|5.4|
> > |||RB-LRA|86.2|5.6|
> # A4) Writing Clarity & Notation Explanation
> >- The matrices $\tilde{U}$ and $\tilde{V}$ in Eq. (5) refer to trainable low-rank residual branches introduced to compensate for information loss during the standard SVD-based LRA.
> >- We provide a theoretical explanation of this setup in our response to Reviewer BTg6 and kindly refer the reviewer to that section for further clarification.
> >- If granted the opportunity to submit a camera-ready version, we will define these notations at their first occurrence and provide a brief explanation of their role and motivation in the Proposed Method section to enhance clarity.
> # A5) How to select the rank(r)?
> >- We determined the rank (r) per layer based on the target model size.
> >- Specifically, we retained only the top singular values after SVD to meet the desired compression ratio.
> >- Since singular values are defined as the square roots of eigenvalues of a positive semi-definite matrix, they are always positive and typically sorted by magnitude for truncation.
> >- The exact rank settings are available in our code, and we will include them in the paper for reproducibility in the camera-ready.
> # Reference
> > [1] Conformer: Convolution-augmented Transformer for Speech Recognition. Interspeech'20

---

> > ### Comment · Reviewer_vjxZ · 2025-04-07
> >
> > Thank you for the detailed rebuttal. All of my concerns have been addressed, I just raised my original score.

---

> > > ### Author Response · Authors · 2025-04-07
> > >
> > > Dear Reviewer,
> > >
> > > Thank you so much for taking the time to carefully read our rebuttal and for reconsidering your evaluation.
> > > We truly appreciate your thoughtful engagement and your updated assessment.
> > >
> > > Your constructive feedback and willingness to re-evaluate played a vital role in helping us improve and clarify our work. We're sincerely grateful for your open-mindedness and for contributing to a fair and productive review process.
> > >
> > > Thank you again for your time and effort.
> > >
> > > Best regards

---

### Official Review · Reviewer_BTg6 · 2025-03-14

**Overall Recommendation:** 3

**Summary:**

The authors propose RB-LRA method which introduces a reparameterizable residual branch to compensate for information loss due to LRA. Weight reconstruction (WR) initializes the residual branch with weights discarded during decomposition, mitigating accuracy loss. To further improve accuracy, the method incorporates feature-based KD. Applying LRA creates large activation outliers, leading to increased quantization errors. So, they propose WADS which identifies layers where weight quantization errors are significant and applies per-token quantization and channel scaling to minimize accuracy loss during quantization. With these, they can achieve significant compression and latency improvements on the Swin-B and DeiT-B models with minimum loss in accuracy.

**Claims And Evidence:**

The paper claims that
1. RB-LRA effectively reduces parameters with minimal accuracy drop
2. Knowledge distillation enhances RB-LRA accuracy
3. WADS effectively reduces Quantization induced accuracy loss
4. Their method achieves significant inference speedups
These claims are supported by experimental evidence

The paper claims to provide a compression solution that applies to ViT models in general, but only the Swin and DeiT models are tested at tiny base sizes. Further evidence on other models such as the ViT model and larger sized models is yet to be seen.

**Essential References Not Discussed:**

PTQ4ViT: Post-training quantization for vision transformers with twin uniform quantization

**Experimental Designs Or Analyses:**

They cover a wide variety of experiments validating their claims across classification, detection, and segmentation tasks. Their ablation studies in the main paper and appendix shine light on the importance of various design choices they make throughout the paper.

**Methods And Evaluation Criteria:**

They test their method across various baselines on the ImageNet classification task and the Detection and Segmentation tasks on the MSCOCO dataset. Inference speeds are tested on Android and on the NVIDIA Jetson Xavier platforms to measure improvements in the real use cases.

**Other Comments Or Suggestions:**

See strengths and weaknesses above

**Other Strengths And Weaknesses:**

Other strengths:
1. The ideas in the paper are clearly presented making it easier for the reader to understand the concepts
2. The paper draws from many methods that boost efficiency in transformers such as LRA, quantization, activation scaling etc.,
3. They provide performance numbers measured on actual devices making their method ready for immediate deployment.
4. They cover a wide variety of baseline methods and ablation studies

Other weaknesses
1. The paper can improve with a theoretical analysis of the design choices they make
2. Testing their algorithm on larger models such as ViT-H, ViT-L etc.,
3. Does their method also work on transformers for Generative modeling such as DiT or is it only restricted to discriminative models?

**Questions For Authors:**

See strengths and weaknesses above

**Relation To Broader Scientific Literature:**

This paper builds on prior research in low-rank approximation (LRA), knowledge distillation (KD), and quantization for Vision Transformers (ViTs) by introducing RB-LRA, a parameterizable branch that mitigates information loss from LRA, and WADS, an LRA-aware quantization method that reduces accuracy degradation from outliers. Unlike existing SVD-based LRA methods (Kumar, 2022; Zhang et al., 2023), which suffer from accuracy drops, RB-LRA reconstructs lost weight information through a residual branch. The paper also improves upon prior KD-based LRA methods (Guo et al., 2024) by introducing block-wise KD and optimizing fine-tuning at the transformer block level. Compared to SmoothQuant (Xiao et al., 2023) and RepQ-ViT (Li et al., 2023), WADS integrates per-token quantization and weight-aware scaling, significantly improving robustness for compressed models. Additionally, the paper demonstrates practical improvements in inference speed (1.9×–3.2× on mobile, 1.5×–2.5× on edge devices), making it a comprehensive and efficient compression framework for real-world deployment.

**Theoretical Claims:**

They claim that the SVD-based initialization of the parameterizable branch reduces information loss compared to LRA. However, no theoretical analysis of this procedure is given.

---

> ### Author Rebuttal · Authors · 2025-03-31
>
> We thank the reviewers for their valuable feedback. Refreshing the page (F5) helps generate equations properly!
>
> # A1) Evaluation of Large-Scale Models
> >- As suggested by the reviewer, we evaluate our framework on ViT-L for large-scale models.
> >- Experimental results show that applying the RB-LRA method improves accuracy by 0.67%.
> >- Furthermore, RB-LRA with WADS achieves 0.18% higher accuracy compared to the baseline while compressing the model by 86.3%, as shown in Table A.
>
> **Table A**
> > |Model|Model Size (MB)|Method|Acc.(%)|
> |-|:-:|-|-|
> |ViT-L|1217.2|Baseline|85.84|
> ||669.2|RB-LRA|86.51|
> ||167.3|RB-LRA + WADS|86.02|
> # A2) Theoretical Analysis
> >- We provide a brief theoretical analysis of RB-LRA and WR in response to the reviewer's comment.
> >- We first define the error based on the Frobenius norm as follows:
> $$W_r = \sum_{i=1}^{r} \sigma_i u_i v_i^{T}, \quad
> \lVert W - W_r \rVert_F^2 =\left\lVert \sum_{i=r+1}^{\min(m,n)} \sigma_i u_i v_i^{T} \right\rVert_F^2 =
> \sum_{i=r+1}^{\min(m,n)} \sigma_i^2 $$
> >- The error in our RB-LRA method is expressed as follows:
> $$\lVert W - (W_r + E) \rVert_F^2 =\left\lVert \sum_{i=r+1}^{\min(m,n)} \sigma_i u_i v_i^{T} - E \right\rVert_F^2$$
> >- Thus, the optimization objective can be formulated as:
> $$\min_{E} \lVert W - (W_r + E) \rVert_F^2$$
> >- While $E$ can be directly computed, its shape matches the original weight, increasing the size from $\mathcal{O}(MN)$ to $\mathcal{O}(2MN)$ and defeating the purpose of low-rank approximation
> >-  We thus design a reparameterizable error matrix using a residual branch:
> $$\tilde{E} = \left( V \tilde{U}^{T} + \tilde{V} U^{'T} + \tilde{V} \tilde{U}^{T} \right)$$
> >- The error matrix guides fine-tuning (FT) toward optimal accuracy.
> >- In the setting of Eq. (5), the initial $\tilde{E}$ matrix with WR applied is expressed as follows:
> $$\tilde{E} = {V}U^{'T} \approx \sum_{i = r+1}^{r+l} \sigma_i u_i v_i^{T}, \quad \text{with } l < \min(m,n) - r
> $$
> >- Finally, the resulting error incorporating this component is expressed as follows:
> $$W_{\text{WR}} = W_r + \tilde{E} \approx \sum_{i = 1}^{r+l} \sigma_i u_i v_i^{T}$$
> >- Consequently, the information loss based on the Frobenius norm for the conventional LRA and the proposed RB-LRA with WR can be compared as follows:
> $$\lVert W - (W_r + \tilde{E}) \rVert_F^2 < \lVert W - W_r \rVert_F^2 = \sum_{i = r+1}^{\min(m,n)} \sigma_i^2$$
> >- Ultimately, the proposed initialization method can be seen as better preserving the information in the discarded parameter dimensions.
> >- We further provide a theoretical analysis of how the proposed method achieves optimal performance through FT.
> >- Discarded singular vectors may retain task-relevant information, leading the gradient to exhibit positive projections onto them during FT.
> >- Based on this observation, it is reasonable to assume a positive correlation between the discarded singular vectors and the gradient.
> >- Accordingly, when $\tilde{V}$ is WR-initialized, it satisfies:
> >$$
> \mathbb{E}\left[\left\langle \tilde{V}, \nabla_W \mathcal{L} \right\rangle_F\right]
> = \sum \sigma \cdot \mathbb{E}\left[\left\langle \nabla_W \mathcal{L}, v \right\rangle\right] > 0
> $$
> >- Whereas random initialization $\tilde{V} \sim \mathcal{N}(0, 1)$  yields:
> $$
> \mathbb{E} \left[\left\langle \tilde{V}, \nabla_W \mathcal{L} \right\rangle_F\right] = \sum \mathbb{E}\left[\tilde{V}\right]\left(\nabla_W \mathcal{L}\right) = 0
> $$
> >- The WR method can be seen as being aligned with the gradient, suggesting that it facilitates faster convergence toward the optimal point in the loss landscape.
> >- The effectiveness of our method is demonstrated in Table 5.
> # A3) Comparision with PTQ4ViT
> >- We compare our method with PTQ4ViT as requested.
> >- As shown in Table B, at a similar compression ratio, our method yields a smaller accuracy drop on DeiT-B, indicating a better trade-off.
>
> **Table B**
> >|Model|Method|Prec.|Model Size (MB)|Acc.(%)|
> |-|-|-|:-:|-|
> |DeiT-B|PTQ4ViT|INT4|43.3|64.30|
> ||Ours|INT8|44.4|**80.56**|
> # A4) Applicability to Diverse Tasks
> >- Although we attempted to evaluate our method on DiT-XL/2, its architecture demands extensive resources and time, making it infeasible within the rebuttal period.
> > Specifically, one epoch takes 40 hours on a single A100 GPU.
> >- As an alternative, we demonstrated the extensibility of our method by applying it to GPT-2.
> > The results can be found in Table E in our response to reviewer vjxZ
> >- We also evaluate performance on a pose estimation task to demonstrate applicability across diverse tasks
> >- Specifically, we apply RB-LRA to the ViTPose [1] model and evaluate its performance on the COCO dataset.
> >- As shown in Table C, we achieve a compression ratio of 25% with less than 1% AP drop, demonstrating the versatility of our method.
>
> **Table C**
> >|Model|Method|Params|AP|AR|
> |-|-|:-:|-|-|
> |ViTPose|Baseline|89.9|75.9|81.0|
> ||RB-LRA|66.8|75.0|80.4|
> # Reference
> >- [1] Vitpose: Simple vision transformer baselines for human pose estimation. NeurlPS'22

---

### Decision · Program_Chairs · 2025-05-01

**Decision:**

Accept (poster)

**Comment:**

The reviewers generally agree that the proposed LRA-QViT method is effective for compressing and quantizing ViTs, with promising results on model efficiency and accuracy. However, concerns remain regarding theoretical analysis and comparisons with state-of-the-art methods. Overall, the paper is weakly accepted with a recommendation to address these issues in the final version if accepted.